# Knowledge, awareness and attitude among practicing oral and maxillofacial surgeons towards tele-dentistry: A cross-sectional survey

Ayse Ozcan-Kucuk ⃝*, Ali Mentes, Adalet Celebi, Fethi Atıl

Department of Oral and Maxillofacial Surgery, Faculty of Dentistry, Mersin University, Mersin, Türkiye

* ayseozcan89@hotmail.com

## Abstract

Tele-dentistry can be defined as the utilization of digital and telecommunication technologies to facilitate remote dental care. This study assessed the knowledge, awareness and practices related to tele-dentistry among oral and maxillofacial surgeons in Türkiye using a cross-sectional survey. A structured questionnaire was distributed to approximately 900 members of the Association of Oral and Maxillofacial Surgery Society (AÇBİD) in Türkiye, and responses from 152 participants were analysed using descriptive statistics and binary logistic regression (p < 0.05). Although 46.1% of respondents reported familiarity with tele-dentistry, only 9.9% had prior experience with it. Overall awareness was moderate and attitudes toward future applications were generally positive; however, actual clinical utilization remained limited. Academic title was the only variable significantly associated with prior awareness and utilization, although the strength of this association was modest and its confidence intervals were wide. None of the predictor variables demonstrated a statistically significant association with the participants' intention to use tele-dentistry in the future. These findings indicate that oral and maxillofacial surgeons in Türkiye are receptive to tele-dentistry but have limited practical exposure. Barriers to implementation may include insufficient training, data security concerns, diagnostic reliability issues, and lack of institutional support. Strengthening specialty training and developing system-level frameworks may support broader adoption in clinical practice. To the best of the authors' knowledge, this is the first study to examine tele-dentistry-related knowledge, awareness, and practices among oral and maxillofacial surgeons in Türkiye.

## Introduction

Telemedicine is defined by the World Health Organization (WHO) as "healthcare services provided using information and communication technologies for the diagnosis, treatment, and prevention of diseases, as well as for research, training of healthcare workers, and improving public health, in situations where distance is important" [1].

**Data availability statement:** All relevant data underlying the findings of this study are available in the Figshare repository at DOI:10.6084/m9.figshare.30709838.

**Funding:** The author(s) received no specific funding for this work.

**Competing interests:** The authors have declared that no competing interests exist.

**Abbreviations:** AÇBİD: Association of Oral and Maxillofacial Surgery Society in Türkiye; WHO: World Health Organization.

With rapid advancement of digital technologies, telemedicine has become increasingly integrated into modern healthcare delivery. One important extension of this concept is tele-dentistry, which utilizes telecommunications and digital tools to facilitate remote consultation, diagnosis, treatment planning, follow-up, and educational activities. By enhancing communication between clinicians and patients, tele-dentistry contributes to improving access to oral healthcare and increasing public awareness of oral and dental health [2].

Tele-dentistry offers several advantages for both patients and healthcare professionals. By enabling remote access to dental care, it supports timely clinical decision-making, reduces practitioner workload, and helps shorten appointment waiting times. Additionally, remote consultations provide a safer alternative for individuals who are bedbound, medically compromised, or unable to attend in-person visits. Collectively, these benefits have contributed to the growing societal acceptance and integration of remote healthcare services [3,4].

Although its acceptance is increasing, the widespread and effective implementation of tele-dentistry largely depends on the readiness of both the public and healthcare professionals. While greater awareness and positive attitudes among the public facilitate acceptance and utilization, it is equally essential that dental professionals—particularly oral and maxillofacial surgeons—possess adequate knowledge, skills, and confidence to apply tele-dentistry effectively. Ensuring professional competence is therefore fundamental for the sustainable and safe integration of tele-dentistry into routine clinical practice [5].

Despite the high prevalence of oral diseases in Türkiye, the implementation of tele-dentistry remains limited. Insufficient technical resources, lack of clear regulations, and inadequate professional training have hindered its development and integration into routine dental care [6]. For tele-dentistry to become both effective and accessible—especially in developing countries such as Türkiye—a well-structured system supported by robust digital infrastructure is essential.

Several international studies have examined the knowledge, awareness, and attitudes of dentists, dental students, and orthodontists toward tele-dentistry [6–12]. Within the field of oral and maxillofacial surgery, existing research has primarily focused on clinical applications, diagnostic accuracy, and the use of tele-dentistry during and after the COVID-19 pandemic, contributing to a better understanding of patient management, access to care, and interprofessional collaboration [13–17]. However, no studies to date have specifically investigated the perspectives of oral and maxillofacial surgeons regarding the use of tele-dentistry. Consequently, their level of knowledge, awareness, and professional attitudes remains largely undocumented, representing a notable gap in the current literature. Therefore, this study aims to evaluate the perceptions, knowledge and practices of oral and maxillofacial surgeons in Türkiye regarding the use of tele-dentistry.

## Methods

### Study design and population

This cross-sectional study was conducted using a structured questionnaire administered between 3 January 2024 to 31 March 2024. The study population consisted

of practicing oral and maxillofacial surgeons in Türkiye. The sample size was calculated using G*Power (Version 3.1.9.6; Heinrich Heine University, Düsseldorf, Germany), with a 95% confidence interval and 95% statistical power and an effect size of $w = 0.411$, based on parameters reported by George et al. [18]. The minimum required sample size was estimated to be 110 participants.

## Ethical approval

Ethical approval was obtained from the Clinical Research Ethics Committee of Mersin University for this study (2023/834). All participants were informed about the purpose and procedures of the study through an information statement provided at the beginning of the questionnaire, and electronic informed consent was obtained from each participant via Google Forms (Google LLC, Mountain View, CA, USA).

## Data collection

For this study, the questionnaire developed by George et al. [18] was adapted and utilized. The modified questionnaire consisting of 26 closed-ended questions was distributed to approximately 900 members of the Association of Oral and Maxillofacial Surgery Society (AÇBİD) in Türkiye via an email list using Google Forms (Google LLC, Mountain View, CA, USA). The first section of the questionnaire collected socio-demographic information, while the second section assessed participants' knowledge, awareness, and attitudes regarding tele-dentistry. Prior to data collection, a pilot study was conducted on 8 oral and maxillofacial surgeons, and revisions were made to improve comprehension, syntax, and clarity.

## Inclusion and exclusion criteria

The inclusion criteria for this study were as follows:

- Participants aged 23 years or older.
- Oral and maxillofacial surgery professionals who agreed to participate in the study (Specialist, Assistant Professor, Associate Professor, Professor).
- Dentists who were currently undergoing specialization or doctoral training in oral and maxillofacial surgery and who agreed to participate (Research Assistants).

  The exclusion criteria for this study were as follows:

- Individuals from dental specialties other than oral and maxillofacial surgery.
- General dental practitioners.
- Individuals who did not provide consent to participate in the study.

## Questionnaire distribution

The quantitative component of the study employed convenience sampling. The initial sampling frame consisted of approximately 900 members registered with AÇBİD. With the support of AÇBİD representatives, the researchers accessed only the registered email addresses of these members; the survey link was then sent directly by the researchers to these addresses. The survey remained open for a total of three months, during which six reminder emails were sent at two-week intervals to non-respondents to increase the response rate. The questionnaire was distributed to research assistants, specialist dentists, assistant professors, associate professors, and professors working in the field of oral and maxillofacial surgery. Participation was entirely anonymous, and no personally identifiable information was collected at any stage.

Participants were able to revise their responses or voluntarily withdraw from the questionnaire at any point prior to submission. No incentives or compensation were provided for completing the survey.

### Study variables

The predictor variables in this study included age, gender, and academic title.

The primary outcome variables were: (1) having heard of tele-dentistry, (2) prior use of tele-dentistry, and (3) intention to use tele-dentistry in the future among oral and maxillofacial surgeons in Türkiye.

"Having heard of tele-dentistry" was defined as prior awareness of the concept (assessed by the item: "Have you heard of tele-dentistry?"). "Prior use of tele-dentistry" referred to the previous experience with tele-dentistry applications in clinical or educational contexts ("Have you used tele-dentistry before?"). "Intention to use tele-dentistry in the future" reflected the participant's likelihood of incorporating tele-dentistry into future clinical practice ("In the future, will you practice tele-dentistry?"). Responses marked as "I don't know" were excluded from the analysis due to the low number of cases.

### Statistical analysis

The study data were analyzed using IBM SPSS Statistics version 23 (SPSS Inc., Chicago, IL., USA) program. Pearson's Chi-Square test was used to examine the categorical variables, while the Bonferroni correction was applied for multiple comparisons. Descriptive results were reported as frequencies and percentages. The associations between predictor variables and outcome variables were assessed using binary logistic regression analysis, and odds ratios (OR) with 95% confidence intervals (95% CI) were calculated. A significance level of $p < 0.05$ was considered statistically significant.

## Results

A total of 152 oral and maxillofacial surgeons from various regions of Türkiye participated in the study. Most respondents were younger surgeons, and a substantial proportion consisted of research assistants (46.1%). The demographic characteristics of the participants are summarized in Table 1.

Overall, the results indicated that although nearly half of the respondents were familiar with tele-dentistry, only a small proportion had previously used it. Participants generally demonstrated a moderate level of awareness and expressed a

**Table 1. Frequency distributions and descriptive characteristics of the participants.**

| Variable | N | % |
|---|---|---|
| Age | | |
| 23-35 | 112 | 73.7 |
| 35-45 | 30 | 19.7 |
| 45-55 | 9 | 5.9 |
| >55 | 1 | 0.7 |
| Gender | | |
| Male | 91 | 59.9 |
| Female | 61 | 40.1 |
| Title | | |
| Research assistant | 70 | 46.1 |
| Specialist | 31 | 20.4 |
| Assistant professor | 20 | 13.1 |
| Associate professor | 19 | 12.5 |
| Professor | 12 | 7.9 |

positive attitude toward the potential future integration of tele-dentistry into oral and maxillofacial surgery practice. The detailed distribution of responses related to knowledge, awareness, and attitudes is presented in Table 2.

Participants who were already familiar with tele-dentistry were significantly more likely to have used it previously, indicating that prior awareness is a key determinant of adoption behavior (p < 0.001; Table 3).

Binary logistic regression analysis indicated that academic title was the only significant predictor of having heard of tele-dentistry. In the multivariate model, participants holding the title of Assistant Professor were nearly four times more likely to have heard of tele-dentistry compared with research assistants (p = 0.018), whereas no significant associations were observed for the other predictor variables (p > 0.05; Table 4).

Similarly, academic title was the only significant factor associated with the prior use of tele-dentistry. Participants holding the title of Professor were markedly more likely to have previously used tele-dentistry compared with research assistants (p = 0.008), while no other variables demonstrated statistically significant associations (p > 0.05; Table 5).

**Table 2. The responses of oral and maxillofacial surgeons' knowledge, practice, attitude to use tele-dentistry.**

| Questions | Yes N (%) | No N (%) | Neutral N (%) |
|---|---|---|---|
| Have you heard of tele-dentistry? | 70 (46.1) | 76 (50) | 6 (3.9) |
| Have you used tele-dentistry before? | 15 (9.9) | 135 (89.4) | 1 (0.7) |
| Does tele-dentistry facilitate access to oral health services during pandemics? | 84 (55.3) | 10 (6.6) | 58 (38.1) |
| Can tele-dentistry be used in oral and maxillofacial surgery? | 64 (42.1) | 25 (16.5) | 63 (41.4) |
| Does tele-dentistry relate to the practice of using computers, internet and technologies to diagnose and advise on treatment remotely? | 103 (67.8) | 2 (1.3) | 47 (30.9) |
| Does tele-dentistry help the patient with consultation and diagnosis in oral and maxillofacial surgery during COVID? | 105 (69.1) | 9 (5.9) | 38 (25) |
| Does tele-dentistry help the patient to share radiographs for diagnosis in oral and maxillofacial surgery? | 106 (70.2) | 4 (2.6) | 41 (27.2) |
| Can tele-dentistry be used in the early detection, diagnosis and treatment of oral diseases? | 95 (62.5) | 13 (8.6) | 44 (28.9) |
| Can tele-dentistry be used in the diagnosis and treatment of patients with temporomandibular disorders? | 64 (42.1) | 42 (27.6) | 46 (30.3) |
| Can the knowledge and experience of other maxillofacial surgeons be utilized during the operation by remote connection in the operations of patients with temporomandibular joint disorders using tele-dentistry? | 84 (55.6) | 20 (13.3) | 47 (31.1) |
| Can implant planning be done using tele-dentistry? | 72 (47.7) | 36 (23.8) | 43 (28.5) |
| Can tele-dentistry be used in the diagnosis and treatment planning of patients with maxillofacial trauma? | 67 (44.1) | 41 (27) | 44 (28.9) |
| Do you think tele-dentistry can be used in patient follow-up after minor oral surgical procedures such as impacted wisdom teeth extraction? | 107 (70.4) | 14 (9.2) | 31 (20.4) |
| Can tele-dentistry be used in the follow-up of MRONJ patients? | 85 (55.9) | 31 (20.4) | 36 (23.7) |
| Can maxillofacial surgeons evaluate patients in virtual meetings with other branches such as ear nose and throat (ENT) specialist and plastic surgery? | 122 (80.8) | 3 (2) | 26 (17.2) |
| Does tele-dentistry help non-specialist dentists to refer their patients to specialist dentists? | 115 (75.7) | 2 (1.3) | 35 (23) |
| Can tele-dentistry be used for training in dentistry and maxillofacial surgery? | 94 (61.8) | 20 (13.2) | 38 (25) |
| Does tele-dentistry help reduce clinical costs in dentistry? | 84 (55.3) | 14 (9.2) | 54 (35.5) |
| Do you think tele-dentistry saves time for dentists and maxillofacial surgeons? | 81 (53.3) | 23 (15.1) | 48 (31.6) |
| Does tele-dentistry facilitate access to patients in underserved areas (e.g., rural areas)? | 98 (64.5) | 13 (8.5) | 41 (27) |
| Can tele-dentistry be used for patient appointment scheduling? | 103 (68.2) | 6 (4) | 42 (27.8) |
| Does tele-dentistry violate patients' personal data and privacy? | 29 (19.1) | 53 (34.9) | 70 (46) |
| Can tele-dentistry be added to routine dental practice? | 80 (52.6) | 20 (13.2) | 52 (34.2) |
| Can tele-dentistry be used in other branches of dentistry other than oral and maxillofacial surgery? | 111 (73.5) | 3 (2) | 37 (24.5) |
| Do you think that examinations using computers, telephones and telecommunication technologies, as in the traditional clinical setting, are correct? | 45 (30.2) | 55 (36.9) | 49 (32.9) |
| In the future, will you practice tele-dentistry? | 75 (49.7) | 18 (11.9) | 58 (38.4) |

Notes: Data are presented as N and %.

**Table 3. The relationship between having heard of tele-dentistry and previous use of tele-dentistry.**

| | Have you heard of tele-dentistry? | | | Test statistics | Effect size | P-value |
|---|---|---|---|---|---|---|
| | Yes N (%) | No N (%) | Neutral N (%) | | | |
| Have you used tele-dentistry before? | | | | | | |
| Yes | 15 (21.7)a | 0 (0)b | 0 (0)ab | 43.93 | 0.381* | **<0.001** |
| No | 54 (78.3)a | 76 (100)b | 5 (83.3)a | | | |
| Neutral | 0 (0)a | 0 (0)a | 1 (16.7)b | | | |

Data are presented as N and %. bold: p<0.05.

Pearson Chi-Square Test; a-b: There is no difference between groups with the same letter, *Cramer's V.

**Table 4. Binary logistic regression analysis of independent variables affecting the likelihood of hearing about tele-dentistry.**

| | Have you heard of tele-dentistry? | | Univariate | | Multiple | |
|---|---|---|---|---|---|---|
| | No N (%) | Yes N (%) | OR (%95 CI) | p-value | OR (%95 CI) | p-value |
| Age | | | | | | |
| 23-35 | 66 (61.1) | 42 (38.9) | Reference | | | |
| 35-45 | 8 (26.7) | 22 (73.3) | 4.32 (1.76–10.6) | **0.001** | 1.93 (0.54–6.89) | 0.309 |
| 45> | 2 (25) | 6 (75) | 4.71 (0.91–24.46) | 0.065 | 1.05 (0.07–16.08) | 0.975 |
| Gender | | | | | | |
| Male | 48 (53.9) | 41 (46.1) | Reference | | | |
| Female | 28 (49.1) | 29 (50.9) | 1.21 (0.62–2.36) | 0.571 | 1.47 (0.7–3.07) | 0.307 |
| Title | | | | | | |
| Research Assistant | 46 (67.6) | 22 (32.4) | Reference | | | |
| Specialist | 17 (58.6) | 12 (41.4) | 1.48 (0.6–3.62) | 0.395 | 1.36 (0.53–3.5) | 0.526 |
| Assistant Professor | 6 (30) | 14 (70) | 4.88 (1.65–14.41) | **0.004** | 3.96 (1.27–12.34) | **0.018** |
| Associate Professor | 5 (27.8) | 13 (72.2) | 5.44 (1.72–17.17) | **0.004** | 3.38 (0.72–15.88) | 0.122 |
| Professor | 2 (18.2) | 9 (81.8) | 9.41 (1.87–47.27) | **0.006** | 8.16 (0.57–116.65) | 0.122 |

Cox&Snell $R^2$=%14.6; Nagelkerke $R^2$=%19.5; Frequency (percentage); OR: Odds Ratio. CI: Confidence Interval. Bold values indicate statistical significance (p<0.05).

Finally, binary logistic regression analysis showed that none of the examined variables significantly predicted the intention to use tele-dentistry in the future. No statistically significant associations were identified in either the univariate or multivariate models (p>0.05; Table 6).

## Discussion

The present study provides new insights into oral and maxillofacial surgeons' knowledge, awareness, and clinical practices regarding tele-dentistry, directly addressing the gap in the literature on how this professional group perceives and utilizes this technology. To the best of our knowledge, this is the first study in Türkiye to focus specifically on the tele-dentistry–related perceptions and behaviors of oral and maxillofacial surgeons, offering a clearer understanding of their current level of readiness for digital health integration. Tele-dentistry plays an important role in patient triage, remote consultations, postoperative monitoring, and interprofessional collaboration [13,15,17]. Consistent with previous reports [6,7,11,18], familiarity with tele-dentistry appears to be increasing; however, notable challenges remain in translating this knowledge into routine clinical use.

**Table 5. Binary logistic regression analysis of independent variables with previous use of tele-dentistry.**

| | Have you used tele-dentistry before? | | Univariate | | Multiple | |
|---|---|---|---|---|---|---|
| | No N (%) | Yes N (%) | OR (%95 CI) | p-value | OR (%95 CI) | p-value |
| **Age** | | | | | | |
| 23-35 | 103 (92) | 9 (8) | Reference | | | |
| 35-45 | 27 (90) | 3 (10) | 1.27 (0.32- 5.02) | 0.732 | 0.11 (0.01–1.98) | 0.135 |
| 45> | 5 (62.5) | 3 (37.5) | 6.87 (1.41- 33.51) | **0.017** | 0.08 (0–3.45) | 0.191 |
| **Gender** | | | | | | |
| Male | 80 (88.9) | 10 (11.1) | Reference | | | |
| Female | 55 (91.7) | 5 (8.3) | 0.73 (0.24–2.25) | 0.580 | 1.06 (0.3–3.81) | 0.926 |
| **Title** | | | | | | |
| Research Assistant | 65 (92.9) | 5 (7.1) | Reference | | | |
| Specialist | 29 (93.5) | 2 (6.5) | 0.9 (0.16–4.89) | 0.900 | 1.1 (0.2–6.07) | 0.913 |
| Assistant Professor | 19 (95) | 1 (5) | 0.68 (0.08–6.22) | 0.736 | 0.94 (0.1–8.77) | 0.959 |
| Associate Professor | 16 (88.9) | 2 (11.1) | 1.63 (0.29–9.15) | 0.582 | 7.46 (0.68–82.28) | 0.101 |
| Professor | 6 (54.5) | 5 (45.5) | 10.83 (2.43–48.32) | **0.002** | 119.57 (3.46–4136.53) | **0.008** |

Cox&Snell $R^2$=%8.8; Nagelkerke $R^2$=%18.5; Frequency (percentage); OR: Odds Ratio. CI: Confidence Interval. Bold values indicate statistical significance ($p < 0.05$).

**Table 6. Binary logistic regression analysis of independent variables affecting the likelihood of considering the use of tele-dentistry in the future.**

| | In the future, will you practice tele-dentistry? | | Univariate | | Multiple | |
|---|---|---|---|---|---|---|
| | No N (%) | Yes N (%) | OR (%95 CI) | p-value | OR (%95 CI) | p-value |
| **Age** | | | | | | |
| 23-35 | 12 (20) | 48 (80) | Reference | | | |
| 35-45 | 2 (8) | 23 (92) | 2.88 (0.59–13.92) | 0.189 | --- | --- |
| 45> | 4 (50) | 4 (50) | 0.25 (0.05–1.15) | 0.075 | --- | --- |
| **Gender** | | | | | | |
| Male | 14 (22.2) | 49 (77.8) | Reference | | | |
| Female | 4 (13.3) | 26 (86.7) | 1.86 (0.56–6.22) | 0.315 | 1.73 (0.49–6.09) | 0.394 |
| **Title** | | | | | | |
| Research Assistant | 6 (17.6) | 28 (82.4) | Reference | | | |
| Specialist | 7 (36.8) | 12 (63.2) | 0.37 (0.1–1.33) | 0.126 | 0.39 (0.11–1.41) | 0.150 |
| Assistant Professor | 1 (8.3) | 11 (91.7) | 2.36 (0.25–21.9) | 0.451 | 2.33 (0.25–21.78) | 0.458 |
| Associate Professor | 1 (5.6) | 17 (94.4) | 3.64 (0.4–32.91) | 0.250 | 3.87 (0.42–35.22) | 0.230 |
| Professor | 3 (30) | 7 (70) | 0.5 (0.1–2.51) | 0.400 | 0.55 (0.11–2.79) | 0.467 |

Cox&Snell $R^2$=%8.9; Nagelkerke $R^2$=%14.2; Frequency (percentage); OR: Odds Ratio. CI: Confidence Interval. Bold values indicate statistical significance ($p < 0.05$).

A clear discrepancy emerged between awareness and actual clinical practice. Despite the fact that many participants were familiar with the concept of tele-dentistry, only a small proportion had incorporated it into routine patient care. Similar patterns observed among general dentists and dental students [6,9,19] indicate that awareness alone is insufficient to drive adoption.

Studies conducted in different regions and populations have reported higher levels of awareness compared with the findings of the present study [11,18,20]. Pradhan et al. reported that postgraduate dental students in India demonstrated relatively high levels of knowledge and awareness regarding tele-dentistry, a pattern attributed to structured educational exposure and the incorporation of tele-dentistry topics into their academic training [11]. Similarly, George et al. found that orthodontists in Kerala exhibited elevated awareness during the COVID-19 pandemic, emphasizing the influence of institutional support and telehealth-driven clinical policies on professional engagement [18]. In Rwanda, Murererehe et al. showed that dental professionals displayed strong awareness of the benefits and applications of tele-dentistry, which the authors linked to supportive system-level conditions within the national healthcare infrastructure [20]. When considered alongside these studies, the lower awareness and limited clinical adoption observed among oral and maxillofacial surgeons in Türkiye suggest that comparable system-level facilitators—such as formal training, telehealth-oriented policies, and institutional integration—may not yet be sufficiently established to promote widespread uptake of tele-dentistry in this professional group.

In the present study, academic titles emerged as a meaningful determinant of tele-dentistry awareness and use. Senior clinicians demonstrated greater engagement than junior practitioners, a pattern that may be attributable to their broader institutional responsibilities, increased exposure to digital health initiatives, and more active involvement in administrative or academic decision-making processes. Similarly, studies conducted in Saudi Arabia and Brazil have shown that dentists with advanced professional standing or longer clinical experience tend to exhibit higher awareness, more favorable attitudes, and greater readiness to engage in tele-dentistry [6,21]. Similar results have been reported in studies from Australia and Sudan, where digital readiness and willingness to engage in tele-dentistry were shown to be shaped by professional role, clinical seniority, and the academic environments in which clinicians practice [14,22]. These findings further support the notion that seniority may facilitate adoption by increasing exposure to technological developments and greater involvement in institutional planning and decision-making processes.

In the present study, a strong association was identified between tele-dentistry awareness and prior use, suggesting that familiarity may facilitate its integration into clinical practice. This relationship is consistent with prior reports indicating that targeted training programs and structured awareness initiatives can significantly enhance tele-dentistry adoption [23]. The higher levels of awareness observed among dental professionals, dental students and trainees in the studies by Akçiçek et al. and Alipour et al. further emphasize the importance of early curricular exposure [24,25]. These findings collectively highlight the importance of systematic education—both at the undergraduate level and through continuing professional development—to facilitate the transition from conceptual understanding to practical implementation. Integrating telehealth-related content into dental curricula and strengthening ongoing training opportunities may therefore help reduce the gap between awareness and routine clinical use.

Despite these findings, no demographic or professional variable was found to significantly predict the future intention to use tele-dentistry. This indicates generally positive attitudes toward digital technologies, yet long-term implementation is dependent on supportive systemic factors such as governmental policies, reimbursement pathways, and digital infrastructure [26]. Evidence from Wood et al., Gangwani et al., and Miranda-Hoover et al. showed that telemedicine use declined after pandemic restrictions eased, underscoring the importance of sustained structural support rather than short-term adaptation [13,15,16].

Participants recognized several advantages of tele-dentistry, including improved access to care, reduced costs, time efficiency, and enhanced collaboration between OMFS and related specialties. However, concerns persisted regarding data security and diagnostic accuracy. Similar issues have been noted in previous studies, which emphasize the need for secure systems and validated diagnostic protocols [27–30]. Strengthening legal frameworks, establishing cybersecurity standards, and developing evidence-based tele-diagnostic guidelines may help improve confidence in tele-dentistry among both clinicians and patients.

## Limitations

This study has several limitations. Although the sample size was statistically adequate (n = 152) and comparable to previous surveys [18], the findings may not be fully generalizable to all OMFS surgeons in Türkiye. Furthermore, the

questionnaire used in this study was adapted from previously published surveys by selecting and combining relevant items. Since it was not subjected to full post-modification psychometric validation (e.g., reliability or validity testing), the measurement precision may be limited.

Second, participant recruitment was based on voluntary responses to a Google Forms link distributed through AÇBİD and the non-response rate for this study could not be determined. This may have introduced selection bias, as individuals with a greater interest in tele-dentistry may have been more inclined to participate. Furthermore, because the number of surgeons who received or viewed the link but did not to respond was unknown, the extent of non-response bias could not be evaluated. As a result, the generalizability of the findings to all OMFS surgeons may be limited. Nonetheless, participation from multiple institutions may have partially mitigated these concerns.

Third, the use of convenience sampling rather than random sampling restricts the external validity of the results. Although random sampling would reduce bias, it was not feasible due to logistical and geographic limitations. Despite this, the findings offer meaningful insight into current trends in tele-dentistry adoption among OMFS surgeons in Türkiye.

From a methodological perspective, the explanatory power of the regression model was modest, and several odds ratios had wide confidence intervals. This suggests that variables such as age and gender may exert limited influence and that unmeasured factors—such as institutional support, training opportunities, and policy frameworks—should be explored in future studies.

Lastly, the cross-sectional design prevents causal inference or the assessment of changes over time. Longitudinal or mixed-method research could better capture how tele-dentistry awareness and behavior evolve. Furthermore, the study focused exclusively on clinicians; including patient perspectives would provide a more comprehensive understanding of tele-dentistry implementation.

Despite these limitations, this study provides valuable baseline data on tele-dentistry in Türkiye and indicates that while individual awareness is growing, sustained integration into OMFS practice will require stronger institutional, educational, and policy-level support.

## Conclusions

In conclusion, oral and maxillofacial surgeons in Türkiye exhibit moderate awareness and generally positive attitudes toward tele-dentistry; however, its practical implementation remains limited. Academic seniority was identified as a key factor influencing both knowledge and experience, while demographic characteristics such as age and gender showed minimal impact. These findings suggest that institutional and policy-level determinants may play a more decisive role in shaping tele-dentistry adoption than individual characteristics. To support systematic and sustainable integration of tele-dentistry into oral and maxillofacial surgical practice, it is essential to incorporate tele-dentistry into postgraduate specialty training, strengthen digital health education, and establish comprehensive national telehealth strategies.

## Author contributions

**Conceptualization:** Ayse Ozcan-Kucuk, Ali Mentes, Adalet Celebi, Fethi Atıl.

**Data curation:** Ayse Ozcan-Kucuk, Ali Mentes, Fethi Atıl.

**Formal analysis:** Ayse Ozcan-Kucuk.

**Funding acquisition:** Ayse Ozcan-Kucuk, Ali Mentes, Adalet Celebi, Fethi Atıl.

**Investigation:** Ayse Ozcan-Kucuk, Ali Mentes, Adalet Celebi, Fethi Atıl.

**Methodology:** Ayse Ozcan-Kucuk.

**Project administration:** Ayse Ozcan-Kucuk, Ali Mentes, Adalet Celebi, Fethi Atıl.

**Resources:** Ayse Ozcan-Kucuk.

**Software:** Ayse Ozcan-Kucuk.

**Supervision:** Ayse Ozcan-Kucuk.

**Validation:** Ayse Ozcan-Kucuk.

**Visualization:** Ayse Ozcan-Kucuk.

**Writing – original draft:** Ayse Ozcan-Kucuk, Ali Mentes, Adalet Celebi, Fethi Atıl.

**Writing – review & editing:** Ayse Ozcan-Kucuk.

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
