## [Decision Letter · Decision Letter 0]

18 Sep 2025

Dear Dr. Ozcan Kucuk,

Thank you for submitting your manuscript to PLOS ONE. After careful consideration, we feel that it has merit but does not fully meet PLOS ONE’s publication criteria as it currently stands. Therefore, we invite you to submit a revised version of the manuscript that addresses the points raised during the review process.

We look forward to receiving your revised manuscript.

Kind regards,

Prita Abhay Dhaimade

Academic Editor

PLOS ONE

Journal Requirements:

2. In the online submission form, you indicated that the data underlying the results presented in the study are available from (Associate Professor Dr. Ayse OZCAN KUCUK E-mail address: ayseozcan89@hotmail.com

Phone number: +90 324 361 00 01).

Reviewers' comments:

Reviewer's Responses to Questions

**Comments to the Author**

1. Is the manuscript technically sound, and do the data support the conclusions?

Reviewer #1: Yes

Reviewer #2: Yes

2. Has the statistical analysis been performed appropriately and rigorously?

Reviewer #1: Yes

Reviewer #2: Yes

3. Have the authors made all data underlying the findings in their manuscript fully available?

Reviewer #1: Yes

Reviewer #2: Yes

4. Is the manuscript presented in an intelligible fashion and written in standard English?

Reviewer #1: Yes

Reviewer #2: Yes

Reviewer #1: Thank you

All comments are added in the manuscript

Please address these

The understanding of a Discussion must especially be addressed

It has a lot of potential and has all the information that is needed

Reviewer #2: 1. Materials and methods

Sampling: convenience sampling method does raise some concerns regarding overall representation. The author should write how the sample size is selected? and calculated?

Questionnaire: While some parts of the study are based on previous research, there is no information on what changes were made, if any, and if their psychometric properties–validation and reliability, post-change–were determined, all these points should be considered and placed on the text.

Analysis. Only Chi-square and Bonferroni tests are referenced. I suggest to add regression analysis, which is very important to assess the independent predictors of awareness/usage more accurately.

Clarify how the non-response rate can affect generalizability.

2. Results/

2.1. The results are fair, they should cover all aspects of the study. The predictors such as age, title and independent variables can be reanalyzed using regression analysis.

2.2. I suggest to add another Tables showing the regression analysis and predictors with crude and adjusted odd ratios.

2.3. Provide confidence intervals (95%) in results for more strong interpretation.

3. Discussion: It is well written; I have some comments:

3.1. Expand the comparison with previous studies on the sample title

3.2. Support the limitations section by highlighting the limitation of low sample size.

.

Reviewer #1: **Yes:**Saadika KhanSaadika KhanSaadika KhanSaadika Khan

Reviewer #2: **Yes:**Moawia Bushra GameraddinMoawia Bushra GameraddinMoawia Bushra GameraddinMoawia Bushra Gameraddin

---

## [Author Response · Author response to Decision Letter 1]

29 Nov 2025

Dear Editor,

We would like to express our sincere gratitude to you and the reviewers for giving us the opportunity to improve our manuscript. Encouraged by your decision letter, we have thoroughly revised the original manuscript and are pleased to submit the updated version for your reconsideration.

The attached document, titled 'POINT BY POINT RESPONSE', provides detailed responses to all reviewer comments, addressing each query individually. In addition, we have uploaded the fully revised manuscript as a separate file titled ‘Revised Manuscript with Track Changes’, in which all modifications are clearly marked in red.

We would be grateful if you could reconsider our revised manuscript for publication in PLOS ONE.

With best regards,

Sincerely yours,

---

## [Decision Letter · Decision Letter 1]

18 Feb 2026

Dear Dr. Ozcan Kucuk,

Thank you for submitting your manuscript to PLOS ONE. After careful consideration, we feel that it has merit but does not fully meet PLOS ONE’s publication criteria as it currently stands. Therefore, we invite you to submit a revised version of the manuscript that addresses the points raised during the review process.

Following peer review, your manuscript is recommended for publication subject to minor revisions. Please address the reviewer comments as detailed in the attached report by reviewer 4.

We look forward to receiving your revised manuscript.

Kind regards,

Prita Abhay Dhaimade

Academic Editor

PLOS One

Journal Requirements:

Reviewers' comments:

Reviewer's Responses to Questions

**Comments to the Author**

Reviewer #3: All comments have been addressed

Reviewer #4: (No Response)

2. Is the manuscript technically sound, and do the data support the conclusions?

Reviewer #3: Yes

Reviewer #4: Yes

3. Has the statistical analysis been performed appropriately and rigorously?

Reviewer #3: Yes

Reviewer #4: Yes

4. Have the authors made all data underlying the findings in their manuscript fully available?

Reviewer #3: Yes

Reviewer #4: Yes

5. Is the manuscript presented in an intelligible fashion and written in standard English?

Reviewer #3: Yes

Reviewer #4: Yes

Reviewer #3: Thank you for your hardwork, The article is quite concise, the methods and result well laid out and the discussion are well-tailored.

Reviewer #4: The manuscript "Knowledge, Awareness and Attitude among Practicing Oral and Maxillofacial Surgeons on Tele-Dentistry: A Cross-Sectional Survey" is a well-written contribution and seems to address a relevant public health issue. It has however some important limitations, mostly related to lack of representativeness of the studied sample. To investigate the perceptions of modern technologies in this group (maxillofacial surgeons), one would have to reach both the digitally competent, younger generation of surgeons, and the older, less technically savvy professionals. Sending an email survey, even with multiple reminders, resulted in a very low response proportion, with responses mostly from those who are willing to complete a questionnaire online, who are most likely more supportive of modern digital technologies. These limitations are correctly addressed in the limitations section. I therefore do not have any major suggestions for the revision, other than recommending a different study design (qualitative study, mixed-methods or a survey with the selection of subjects and data collection method better adapted to the target population).

Minor issues:

- The title is difficult to understand. Maybe the authors could consider replacing "on" to "towards" - this will make it easier to understand what attitudes were investigated.

- Line 116 - "This cross-sectional study" (remove "a")

- Line 168 - In the word "Intention" there seem to be a Turkish letter.

- Line 182 - Instead "The majority of", please use "Most"

- Line 183 - The % should be placed after the number

.

Reviewer #3: No

Reviewer #4: No

---

## [Author Response · Author response to Decision Letter 2]

21 Feb 2026

Manuscript Title: Knowledge, Awareness and Attitude among Practicing Oral and Maxillofacial Surgeons on Tele-Dentistry: A Cross-Sectional Survey

(Revised Manuscript Title: Knowledge, Awareness and Attitude among Practicing Oral and Maxillofacial Surgeons towards Tele-Dentistry: A Cross-Sectional Survey)

Manuscript Number: PONE-D-25-32213

Minor Revision

Reviewer 3

Comments

Thank you for your hardwork, The article is quite concise, the methods and result well laid out and the discussion are well-tailored.

Our response: We sincerely thank the reviewer for their constructive and encouraging feedback. We are grateful for their positive evaluation of the clarity and organization of our methodology, results, and discussion sections.

Reviewer 4

Comments

The manuscript "Knowledge, Awareness and Attitude among Practicing Oral and Maxillofacial Surgeons on Tele-Dentistry: A Cross-Sectional Survey" is a well-written contribution and seems to address a relevant public health issue. It has however some important limitations, mostly related to lack of representativeness of the studied sample. To investigate the perceptions of modern technologies in this group (maxillofacial surgeons), one would have to reach both the digitally competent, younger generation of surgeons, and the older, less technically savvy professionals. Sending an email survey, even with multiple reminders, resulted in a very low response proportion, with responses mostly from those who are willing to complete a questionnaire online, who are most likely more supportive of modern digital technologies. These limitations are correctly addressed in the limitations section. I therefore do not have any major suggestions for the revision, other than recommending a different study design (qualitative study, mixed-methods or a survey with the selection of subjects and data collection method better adapted to the target population).

Minor issues:

- The title is difficult to understand. Maybe the authors could consider replacing "on" to "towards" - this will make it easier to understand what attitudes were investigated.

- Line 116 - "This cross-sectional study" (remove "a")

- Line 168 - In the word "Intention" there seem to be a Turkish letter.

- Line 182 - Instead "The majority of", please use "Most"

- Line 183 - The % should be placed after the number

Our response: We sincerely thank the reviewer for their positive and constructive evaluation of our manuscript. We fully agree with the concerns raised regarding the representativeness of the sample and the potential response bias inherent to email-based surveys. As appropriately noted, these limitations have been addressed in the limitations section of the manuscript. We also appreciate the suggestion regarding alternative study designs (qualitative or mixed-methods approaches), which we will carefully consider in our future research.

Minor Issues:

Comment 1 (Title): The title is difficult to understand. Maybe the authors could consider replacing "on" to "towards".

Our response: Thank you for this helpful suggestion. The title has been revised accordingly and now reads:

"Knowledge, Awareness and Attitude among Practicing Oral and Maxillofacial Surgeons towards Tele-Dentistry: A Cross-Sectional Survey."

Comment 2 (Line 116): "This cross-sectional study" (remove "a").

Our response: We thank the reviewer for this careful observation. The extra "a" has been removed.

Comment 3 (Line 168): In the word "Intention" there seem to be a Turkish letter.

Our response: We apologize for this typographical error. The Turkish character (İ) has been replaced with the standard English letter "I".

Comment 4 (Line 182): Instead "The majority of", please use "Most".

Our response: Thank you for this helpful suggestion. “The majority of” has been replaced with “Most” as recommended.

Comment 5 (Line 183): The % should be placed after the number.

Our response: Thank you for pointing this out. The percentage formatting has been corrected throughout the manuscript.

---

## [Decision Letter · Decision Letter 2]

22 Mar 2026

Knowledge, Awareness and Attitude among Practicing Oral and Maxillofacial Surgeons towards Tele-Dentistry: A Cross-Sectional Survey

PONE-D-25-32213R2

Dear Dr. Ozcan Kucuk,

We’re pleased to inform you that your manuscript has been judged scientifically suitable for publication and will be formally accepted for publication once it meets all outstanding technical requirements.

Kind regards,

Prita Abhay Dhaimade

Academic Editor

PLOS One

Additional Editor Comments (optional):

Reviewers' comments:

Reviewer's Responses to Questions

**Comments to the Author**

Reviewer #4: All comments have been addressed

2. Is the manuscript technically sound, and do the data support the conclusions?

Reviewer #4: Yes

3. Has the statistical analysis been performed appropriately and rigorously?

Reviewer #4: Yes

4. Have the authors made all data underlying the findings in their manuscript fully available?

Reviewer #4: Yes

5. Is the manuscript presented in an intelligible fashion and written in standard English?

Reviewer #4: Yes

Reviewer #4: My few comments have been addressed and the text is clear. Congratulations on the well written manuscript.

.

Reviewer #4: No

---

## [Editor Report · Acceptance letter]

PONE-D-25-32213R2

PLOS One

Dear Dr. Ozcan Kucuk,

I'm pleased to inform you that your manuscript has been deemed suitable for publication in PLOS One. Congratulations! Your manuscript is now being handed over to our production team.

Kind regards,

on behalf of

Dr. Prita Abhay Dhaimade

Academic Editor

PLOS One